# The Compensation for Losses to Indigenous Peoples Due to the Arctic Industrial Development in Benefit Sharing Paradigm

**Violetta Gassiy** [1,*] and **Ivan Potravny** [2]

[1]   Public administration department, Kuban State University, 350040 Krasnodar, Russia
[2]   Project and Programmes Management Department, Plekhanov Russian State University of Economics, 117997 Moscow, Russia; ecoaudit@bk.ru
*   Correspondence: vgassiy@mail.ru; Tel.: +7-918-438-11-28

**Abstract:** This article discusses the results of research on the benefit sharing system in Russia focusing on compensation of losses to indigenous peoples due to industrial development in the Arctic. The authors analyzed a Russian case-study on the economic mechanisms of coordination and harmonization of multi-vector and conflicting interests in the process of industrial development of traditional lands. The developed recommendations will allow, on the one hand, compensating the losses of the indigenous communities, and, on the other hand, to engage indigenous peoples in the process of environmental management and socio-economic development of their territories. The object of the research was the Republic of Sakha and the indigenous communities of the remote Anabar region. The calculation of losses was considered. The authors suggest using this tool for the traditional lands development, because it helps to define fair compensation due to project impacts and to form a fund for sustainable community development. The considered project was exploring and extracting placer diamonds in Polovinnaya River in Yakutia. This paper also presents the social poll results organized in the indigenous communities in 2017. The results helped to formulate the recommendations for the business on benefit sharing agreements with Anabar communities.

**Keywords:** benefit sharing; indigenous peoples; investment projects; traditional nature management; economic mechanism; losses; compensation; the Arctic; Russia; Yakutia

## 1. Introduction

Currently, large-scale investment projects are being implemented in the Russian Arctic to develop eight core zones: Kola, Arkhangelsk, Nenets, Vorkuta, Yamalo-Nenets, Taimyr-Turukhanskiy, Northern Yakutia and Chukotka [1,2]. Some Russian regions located in the north are the most important strategic territories in terms of natural resources development, exploration and extraction of many minerals (hydrocarbons, gold, silver, diamonds, platinum, ferrous and non-ferrous metals, rare earth raw materials, etc.). At the same time, the intensive development of these territories is often accompanied by the impact on the traditional lands of indigenous peoples and is in contradiction with their way of life and traditional crafts, complicating their livelihoods, including the land withdrawing process used by indigenous communities. These projects are largely related to the exploration and mining of raw materials, transport and economic infrastructure development or military security reasons. The investment projects implementation on core zones development in the Russian Arctic may affect the territories of traditional nature use and influence the traditional lands of the indigenous peoples of the North. Insufficient consideration of the environmental and ethnological component in the justification and implementation of such investment projects in the territories of traditional residence

and traditional economic activities of indigenous peoples can lead to conflicts [3]. The purpose of the article is to improve theoretical approaches and methods, currently used in Russia, for estimating possible losses of indigenous peoples that arise from the impact of exploration and mining of mineral resources in their traditional territories.

The governmental concept of sustainable development of indigenous peoples of the North, Siberia and the Far East of the Russian Federation (2009) was adopted for the development and preservation the traditional lands and traditional nature use. Although the Federal Law of 30 April 1999 No.82-FZ "On the Guarantees of the Rights of the Indigenous Minorities of the Russian Federation" established the legal basis for the rights of the original socio-economic and cultural development of the indigenous minorities, the protection of their traditional lands, traditional lifestyle, management and pastures, the current legislation does not regulate the procedure for losses' compensation due to business economic activities on the traditional lands [4]. We discuss the development and application at the federal level of the losses' compensation methods caused to indigenous peoples and their communities by business organizations of all types of activities. Under these conditions, an important scientific and practical task is the development of effective management tools and mechanisms for regulating and harmonizing relations between business and indigenous peoples during the industrial development of the Arctic.

The Republic of Sakha (Yakutia) is a raw material region, with mining as the leading industry, which is accompanied by negative environmental and social consequences. Indigenous peoples of the North can engage in economic activities only on undisturbed or minimally disturbed lands. The deterioration of the environment and the reduction of reserves of renewable resources has a devastating impact not only on traditional types of environmental management, but also on their mentality, culture and traditions. From this point of view, one of the most acute problems for indigenous peoples is the question of fair compensation for the damage caused by industrial enterprises to territories of traditional nature use, which in turn cause losses to indigenous communities. To solve this problem, it is necessary to develop a methodology for estimating losses of landowners and creating economic mechanisms for the sustainable development of traditional environmental management areas in the context of investment projects in the Arctic. On the other hand, the law does not regulate the procedure for compensation for losses from the economic activities of organizations in the places of traditional residence and traditional economic activities of indigenous peoples [4]. According to Indigenous and Tribal Peoples Convention, the handicrafts, rural- and community-based industries, and subsistence economy and traditional activities of the peoples concerned, such as hunting, fishing, trapping and gathering, shall be recognized as important factors in the maintenance of their cultures and in their economic self-reliance and development. Governments shall, with the participation of these people and whenever appropriate, ensure that these activities are strengthened and promoted [5]. Compensation is considered as a payment for damages that have been caused by an industrial object to indigenous peoples. Compensation from a subsoil user can be monetary (payments to a community member or tribal community) and non-monetary (construction of social or transport infrastructure, implementation of projects for the preservation of cultural heritage, etc.). To ensure the implementation of the Convention, the state is obliged to improve the mechanisms that allow indigenous communities to maintain their handicrafts, receive compensation as a form of economic guarantees of their rights. In Russia, the economic rights of the indigenous peoples are still developing due to the failure of legislation.

For example, in Russia in 2018, two federal legislative acts came into conflict: the Federal Law "On Guarantees of the Rights of the Indigenous Minorities of the Russian Federation" and the Land Code of the Russian Federation. On the one hand, the federal law guarantees small peoples and their associations the free use of lands of various categories in places of traditional residence and traditional economic activities. However, in accordance with the Land Code of the Russian Federation, to persons belonging to the indigenous peoples of the North and their communities, land plots are provided for free use only to accommodate buildings and structures necessary for the preservation and development of traditional lifestyles, business and crafts. As professor A. Sleptsov explained "land

plots from agricultural lands in state or municipal ownership may be transferred to the indigenous peoples' communities for agricultural production, preservation and development of the traditional way of life, management and crafts of the indigenous peoples for rent. At the same time the purchase of the leased land plot into the private property is not allowed" [6]. According to the Land Code from 1 January 2018, the communities of indigenous peoples need to pay rent. This concerns those farms that did not manage to register their land plots for free use in the federal land cadaster and were forced to arrange deer and hunting lands for rent, as well as for newly formed farms of the indigenous peoples. Since, in the Arctic, reindeer pastures and hunting lands occupy huge areas, these rental payments amount to millions of rubles. This conflict in Russian legislation requires speedy resolution; at present, amendments to the Land Code are being considered in the federal parliament. Thus, issues of land rights of indigenous communities and the compensation for losses due to industrial development for damage caused to the territories of traditional nature use are interrelated.

The difficulty of developing a methodology for estimating and compensating for the losses of indigenous peoples in the context of Arctic industrial development also consists in the lack of unity of theoretical approaches and conceptual apparatus in solving this problem. For example, there are studies about the losses of traditional economic activities in the zone of reindeer herding [7] as well as the economic assessment of the damage caused to Arctic ecosystems during the development of oil and gas and other natural resources or the damage caused to traditional environmental management [8,9]. In Russian scientific literature, there is term definition of losses of land users, owners during extraction of minerals in the Russian Arctic, but not the losses of indigenous communities [10]. In our opinion, during the development of the methodology and mechanisms for sustainable environmental management in the Arctic zone, we should talk about compensation of lost profits and losses to the indigenous peoples of the North, their associations, and tribal communities as a result of the implementation of certain industrial projects [11].

## 2. The Study Area

The object of the research was the indigenous communities of the Anabar Dolgan-Even national region, Republic of Sakha (Yakutia). The region is located in the extreme northwest of the Republic of Sakha (Yakutia), between the 71st and 76th degrees of northern latitude. It occupies a vast area of 55.6 thousand km$^2$ and borders in the north with the Laptev Sea. More than 3400 people live in the Anabar region. The share of indigenous peoples of the North in the population of the area is 37%.

Only herding of semi-domesticated reindeer, hunting wild reindeer and white fox, and fishing are of economic importance here. Currently, there are two settlements in the region. The village of Saskylakh is the district center of the Anabar region, where the Evens live. The village of Urung Khaya is the only settlement in the Republic where compactly live the Dolgans, Figure 1.

The Anabar River flows through the territory of the region, one of the largest rivers in the Arctic Ocean basin. The organization of river navigation has created the prerequisites for the development of the productive forces of the vast Anabar region. The region has diamond deposits, hydrocarbon raw materials, and brown coal. Historically, the main economic activity of the Anabar area is agricultural production. Traditionally, the population of the region is engaged in reindeer herding, hunting, and fishing. The most important condition for the growth of the regional economy and the welfare of the population is the development of the mining industry. Today, three diamond-mining enterprises successfully work in the territory owned by Alrosa. The area of industrial enterprise activity for the mineral wealth development expands every year. The municipality has long-term cooperation agreements with all diamond-mining enterprises. Every year, about 150 people from the local communities are enrolled in seasonal diamond mining enterprises.

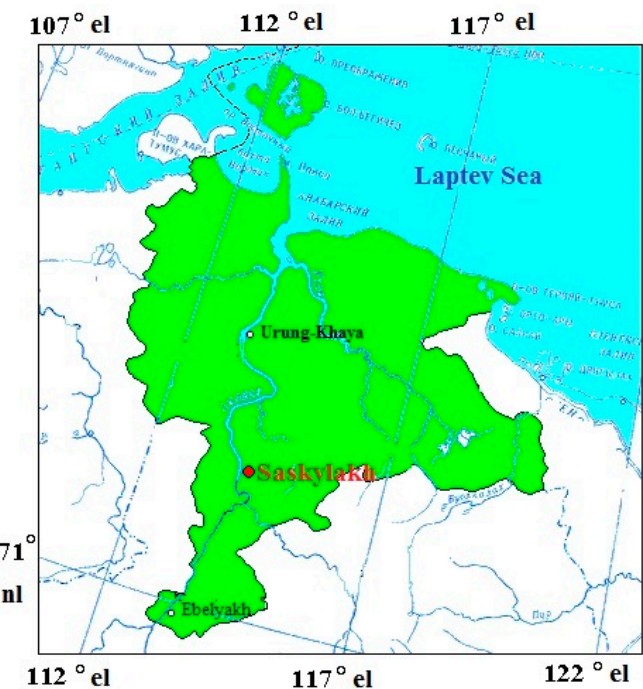

**Figure 1.** Anabar Dolgan-Even national region, Republic of Sakha (Yakutia): the study area (Source: https://web.archive.org/web/20090425140901/; http://www.sitc.ru/monitoring/anabar/index.shtml).

The ethno-social composition of the Anabar region is represented mainly by two groups: the Evens and the Dolgans. The socio-economic situation in the Anabar region is ambiguous. In the region, there is a high birth rate, unlike other northern regions of Yakutia. However, in the region, there is an increase in the migration loss of the population, an outflow of labor resources due to the tense situation on the labor market, Figure 2.

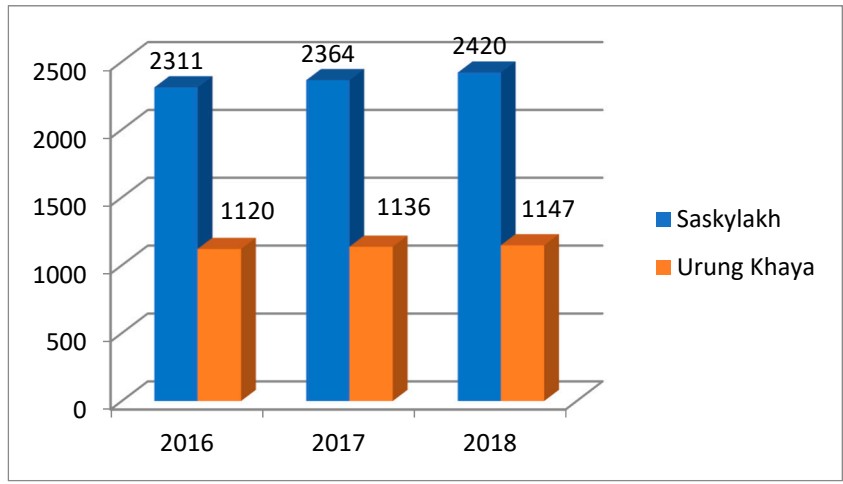

**Figure 2.** The population of Anabar region, Yakutia.

The main causes of adult mortality are infectious and parasitic diseases of neoplasms, diseases of the circulatory system, diseases of the respiratory organs, diseases of the digestive organs, and external causes. Most deaths among external causes are due to alcohol poisoning and suicides [12]. The main cause could be connected with poor economic conditions and high degree of unemployment in the region. In the Soviet period, the handicraft activities were engaged in the public enterprises, and the employment rate was up to 100%. Nowadays, the traditional activities decrease due to low income in this sector and youth outflows.

The sectoral structure of the economy is represented by the following elements:

1. Agriculture and processing of agricultural products
2. Fisheries
3. Personal part-time farm
4. Entrepreneurship
5. Transportation
6. Communication
7. Trade and consumer market

The analysis of statistical data showed that at present the sectoral structure of the Anabar regional economy has narrowed considerably but the big industry is diamond mining. Branches of traditional management are the basis of the tribal communities and municipal enterprises of the district. In the region, 10% of the reindeer population of Yakutia are concentrated. It increases annually (+11% in 2018 compared to 2017). Reindeer carrying capacity is 18,000. At present, in Anabar region, there are 15,409 reindeers managed by eight reindeer herding communities.

In 2017, during the Arctic research expedition, a social survey was organized by the authors in both Anabar villages to study the behavioral and social attitudes of local residents to the implementation of a mineral exploration project on the Polovinnaya River with the following objectives, Figure 3:

- Identify the attitudes of local residents to the socio-economic and environmental problems of the Anabar region to develop recommendations for improving the quality of life in the area.
- Identify the most promising areas of development of the area.
- Identify the correlation of age and other socio-demographic indicators of the population with the perception of the socio-economic and environmental problems of the area, as well as economic activities for the extraction of minerals.
- Identify the attitude of local residents to the economic activities of companies in the exploration and mining of minerals.
- Determine the possible compensation format for conducting mining operations in the district.
- Analyze the needs and attitudes of local residents, which must be considered by companies when carrying out economic activities for the extraction of minerals in the license area.

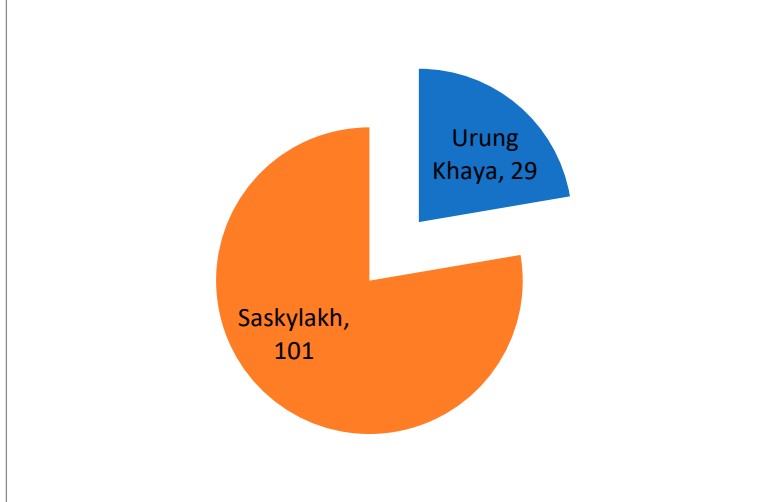

**Figure 3.** Respondents in Saskylakh and Urung Khaya villages (total number of respondents: 130).

The Socio-demographic characteristics of the respondents, Table 1

- Men—59 people (45%).
- Women—71 people (55%).

Representatives of indigenous peoples:

- Evenki—43 people (33%).
- Dolgans—71 people (55%).

Family status

- Married—63.1%.
- Not married—22.3%.
- Divorced—6.9%.
- Widowed—7.7%.

Number of children among respondents

- No—11.5%.
- 1 child—13.8%.
- 2 children—52.3%.
- 3 or more children—22.3%.

**Table 1.** Structure of respondents by type of activity.

| Type of Activity | Respondents, ppl | Share of the Total Number of Respondents % |
|---|---|---|
| Employed | 86 | 66.2 |
| Unemployed | 12 | 9.2 |
| Temporarily unemployed | 11 | 8.5 |
| Retiree | 14 | 10.8 |
| Housewife | 2 | 1.5 |
| Student | 4 | 3.1 |
| Other | 1 | 0.8 |
| Total | 130 | 100.0 |

Residents of the district are most concerned about high food prices (22.5%), while respondents note that the problem lies not only in high prices, but also in the inaccessibility of essential foodstuffs. In a qualitative analysis of the respondents' answers, there is a lack of fresh vegetables and fruits. Note that the amount of monthly income does not affect the degree of perception of this problem, which emphasizes the objectivity of the problem. Respondents (about 5% of all respondents) also noted such a problem as overcharging of air tickets. Other most acute problems of the district are the lack of jobs and low incomes; these problems are especially marked by respondents aged from 30 to 50 years, Figure 4.

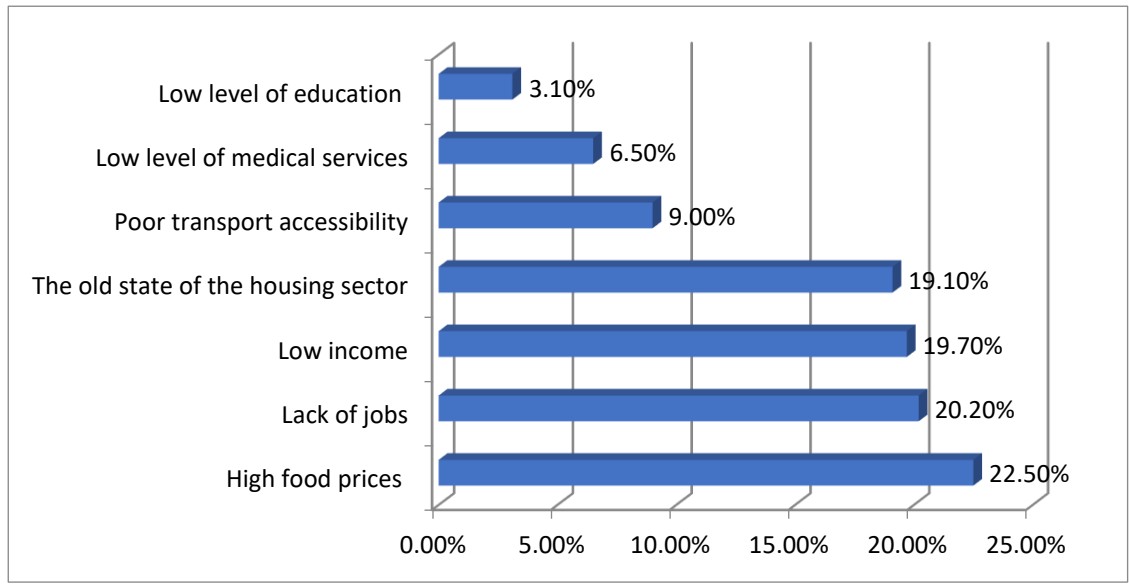

**Figure 4.** Socio-economic problems that concern residents of the area.

Assessing the social and cultural problems of the development of the territory, residents first note such negative phenomena as an increase in morbidity and mortality (20.7%), alcoholism (18.3%) and a loss of communication between people and their culture and traditions (18.3%). It is important to understand that the process of increasing morbidity and mortality is directly related to the quality of medical services, but the respondents do not single out aspects of the quality of medical services as the most significant problems, which can be explained by the low level of general understanding of human health culture. In addition, respondents note that the increase in the number of deaths is due to the lack of infrastructure designed for local climatic conditions. Thus, it is noted that there are no conditions for the conduct of pregnancy and childbirth, while low transport accessibility leads to the death of people in the tundra. Answers of respondents were distributed almost evenly among all the suggested answers, which indicates the high importance of the whole spectrum of the presented problems.

Residents consider the reduction of the number of deer and the change in the ways of their migration to be the most significant environmental problem. The respondents expressed particular concern regarding the quality of the groundwater and the decrease in water in the Anabar River. The results of sociological studies also show that the problem of the export of scrap metal and increased background radiation in the district is particularly acute.

Traditional handicrafts are popular with men aged 40–50 years. However, the attitude and assessment of traditional handicrafts among the local population is very different and controversial. About 14% of respondents noted that income from traditional crafts is the only source of income. This suggests that the occupation of traditional crafts is not the dominant source of income. Moreover, respondents engaged in traditional crafts consider themselves to be unemployed, and, therefore, ready for other types of activity and willing to be employed. On average, income from traditional industries among the population ranges 20–30 thousand rubles, which is on the same level as income from other fields of activity. About 20% of respondents in addition to their regular work are engaged in traditional crafts, the majority of whom (16%) are engaged in fishing. Moreover, such additional employment does not bring income: only about 30% of respondents say that extra employment brings additional earnings. Respondents shared willingly that one of the types of such earnings is the collection of mammoth tusks. Many have pointed out that this work is hard and does not bring a stable level of income and is associated with a risk to life. Describing the occupation of traditional crafts, it should be noted a downward trend in the proportion of the population engaged in this type of activity in Anabar region. These data may also indicate readiness for other types of work.

For the preservation and further development of the traditional sectors of the North, the material and technical base of farms need to be improved, as do the food supply and the social and living conditions of reindeer herding families and their production conditions. One of the important tasks at the present stage of traditional industry development is the creation of conditions for the procurement, processing and marketing of products of traditional sectors. It is necessary to create conditions for the transportation of products to the point of sale. The benefit sharing agreements with subsoil users and business participation in the socio-economic development of the municipal district through projects on traditional culture support are one of the mechanisms for overcoming the crisis of the indigenous communities in Yakutia.

## 3. Concepts and Methods

The peculiarity of the benefit sharing mechanism during industrial development of the Arctic territories abroad supposes the participation of indigenous peoples in the distribution of profits of mining companies. In Russia, business and local communities interact based on compensation payments and agreements on the socio-economic development of their territory. For example, in the United States and Canada, such a mechanism for the distribution of benefits in the interaction of business and indigenous peoples is usually reduced to the problem of the distribution of financial benefits (profits) with the participation of local communities. Scientists consider the benefit sharing system and community agreements with the subsoil-users urgent. Ciaran O'Faircheallaigh suggested implementing community development agreements as a very important tool, especially in the case of mining industries when local communities are affected by the direct impacts of the industrial objects (environmental pollutions, social threats, etc.). As a rule, environmental and social costs incur in the territory developed by the companies while the profit concentrates in other places. This leads to conflicts between local communities and business [13]. In Canada, the exploration and extraction of mineral resources is currently increasing significantly and this makes government and First Nation communities pay more attention to the responsibility of mining companies as they expand their economic activity into Treaty and Aboriginal lands. In most countries, as a rule, the benefit sharing agreements are confidential and concluded between mining companies and indigenous communities. They are specific for certain project and contain the list of the socio-economic and environmental benefits and responsibilities. During such negotiations, the methods to address local needs and involvement mechanics of tribal communities are the focus of many researchers [14]. Brad Gilmour and Bruce Mellett believed that a contract should be preceded by a deep analysis of potential areas of community life that may be adversely affected by an investment project. In addition, the benefit sharing agreement must address future opportunities that could arise during project realization [15]. Tysiachniouk and Tulaeva considered that, in Russian regions, there are various practices of concluding benefit sharing agreements: the peculiarities of the formation of various models of agreements on the distribution of benefits between oil and gas companies and indigenous peoples should be identified. It is important to understand what is significant in the process of their formation: regional specificity, dependence on international actors, features of corporate policies, the level of self-organization of local communities [16]. However, partnership agreements between subsoil users and indigenous communities should provide as compensation for the land fund use, and direct participation in socio-economic development of the traditional lands. One of the tasks of the local authorities in addition to ensuring the interests of the territory and its inhabitants in relations with the company subsoil user could be the motive and stimulating the population to engage in economic activity using the received compensation payments [17].

Analysis of literary sources and economic practice in the Arctic zone allows us to identify the following areas of wealth distribution during industrial development of the territory:

- Compensation: Indemnification of indigenous peoples of the North and their communities, whose activities decrease due to the project.

- Employment: Vocational training and individual employment of representatives of the indigenous peoples of the North, including in planned projects.
- Partnership and cooperation: Mutually beneficial cooperation of the company and indigenous communities through procurement the products of traditional nature use, traditional crafts.
- Co-management: The inclusion of representatives of the indigenous communities in councils to co-manage the project for industrial development of the territory and the interaction of all stakeholders and other areas.

These forms of benefit sharing, in our opinion, can be complemented by business assistance in processing of traditional products, financing and transferring technologies for the agricultural production, subsidizing the communities for traditional economy development [8,13]. Traditional economy provides an environmentally oriented labor market development—green jobs in environmentally friendly production (meat, fish, berries, herbs, etc.) [6].

In Russia, the benefit sharing agreements can be realized as the Program on mitigation of the negative impact of the project (hereinafter—the Program). Negative impact supposes the changes in the traditional lifestyle of indigenous peoples, the need for adaptation and sustainable socio-economic development of local residents in the context of changes arising during the project. Such program may include:

- contracts for compensation for losses to indigenous peoples and traditional lands;
- employment agreements for the indigenous involvement to project development;
- traditional product procurement;
- traditional cultural and environment conservation (financing of traditional holidays etc.).

For example, in Yakutia, the authors became the initiators of the Program realized by Almazy Anabara JSC (Alrosa) and Arctic Capital LLC. The Program includes contracts with indigenous communities and individual entrepreneurs on procurement of agricultural products (fish, meat, etc.), and construction social and transport infrastructure [13]. Such experience should be practiced in Russia by all companies having projects in the Arctic.

*Benefit Sharing Concept in Russia*

The problem of protecting the economic rights of indigenous peoples is related to the fact that, in the Land Code of Russia, indigenous communities are not listed as subjects of law. Therefore, the transfer of land to them by law is not provided. Indigenous communities are forced to register as legal entities and then prove the right to land. However, if, for example, the indigenous peoples live in the forest, then the transfer of land is excluded, since the forest is federal property under Russian law. These and other conflicts in land legislation cause problems with the protection of economic rights, and therefore with the damage assessment to indigenous communities due to negative impact of industrial project as well as the compensation. Many indigenous peoples, Koryaks and Evenks, Nenets and Dolgans, during 2005–2006 lost their own national territories and autonomous districts, and they were united with other regions of the Russian Federation. It was a period of enlargement of the territories due to the unification. As a result, there was a derogation of the rights of indigenous peoples, including difficulties related to formal registration rights on traditional environmental management.

In Russia, the underdevelopment of the legislative system for the protection of the economic rights of indigenous peoples, including the distribution of benefits, means that, in a given region, the practice of interaction between indigenous communities and business is different. For example, in the Krasnoyarsk region, the damage to traditional lands is the result of negotiations. The local government does not use any methodology to calculate objective losses of indigenous communities and the traditional nature use. In the Yamalo-Nenets Autonomous District, to estimate losses and compensate indigenous peoples, it is proposed to proceed from the average income per representative of indigenous peoples living in the project's area of influence. Such an approach does not link to the real damage caused to natural resources since the real income of the local population is low.

Therefore, the aim of the study was to summarize and develop approaches to estimate losses to the indigenous peoples. This was based on the experience using this mechanism in the Republic of Sakha (Yakutia) during 2011–2018 in the framework of conducting the ethnological expertise.

As for the terminological apparatus in Russia, the *ethnological expertise* of the project was a scientific study of the impact of changes in the indigenous lands of indigenous peoples and the socio-cultural development of an ethnic group. At present, the Republic of Sakha is the one region of Russia where law on ethnological expertise has been adopted. The act was adopted on 14 April 2010.

The ethnological expertise of the project should determine the degree of admissibility of a project in the territories that affect the places of residence and traditional economic activities of indigenous peoples. It should also determine possible losses of indigenous peoples.

Such an approach can be considered as one of the elements within the framework of the concept of distribution of benefits during the development of the territory in the Arctic.

Among the mechanisms that support traditional livelihoods of indigenous peoples in Russia, we can distinguish the procurement of products (fish, wild reindeer meat, berries, etc.) by the company, as well as the allocation of targeted grants to support traditional crafts and ethnic development. Today, in most benefit sharing cases, compensation payments prevail. During the Soviet period of the Arctic development, the state bore the brunt of responsibility and costs for the socio-economic development of the territory.

The research in Russia and abroad has identified the following types of benefit sharing in Arctic industrial development:

- Paternalism: The state is responsible for the distribution of benefits, and assumes the main functions for the development of the territory. This type of regulation of environmental management has developed, for example, in Alaska, the United States.
- Social responsibility of the company: The mining companies play an important role in the development of traditional nature use areas, and act as the main carriers of goods and distribute them. An example of such companies is Arctic Capital LLC or Almazy Anabara JSC in Yakutia. They engage in mining of placer diamonds and gold in the Arctic regions [13].
- Partnership: This type of interaction has developed on Sakhalin. The public–private partnership realized between a company and local communities aims to distribute the benefits during the natural gas production on the shelf. This also applies to the benefit sharing system in Canada.
- Contract system for distribution of benefits and traditional crafts support: In this case, the main role in benefit sharing belongs to non-governmental organizations that carry out the economic and non-material assistance individually for indigenous peoples, for their families, tribal communities and other groups.
- The shareholder model: This system supposes that indigenous peoples realize their rights as the owners of shares. Such form of the interaction between indigenous peoples and investors has been developed abroad (Australia, USA, and Canada). In Russia, for example, in the Arctic territories of the Yakutia, some indigenous communities express interest in implementing the shareholder model.

In Russia, due to legislative conflicts and the imperfection of the land ownership system of indigenous peoples, the implementation of a shareholder system is difficult, but the practice of implementing the contract system and partnership, as well as social responsibility, is well developed. However, in each case listed, the basis of negotiations on measures to support the local population should be clearly described as well as potential losses, damages and risks to local communities. At present, the calculation of losses for compensation is currently the main mechanism for a fair negotiation process between indigenous peoples and the investors.

Traditional nature use depends on the natural resource base of the territories determined by the climatic conditions. The main criterion is the natural-ecological differentiation of the territory according to the nature of the distribution of vegetation:

- Integral indicator of physical-geographical features of the territories; and
- Sustainability of natural landscapes to anthropogenic impacts. For example, there are 59 territories of traditional nature use in Yakutia [16].

The authors' concept of calculating losses to indigenous peoples was based on the methodical approach of the resource assessment of the territory. This considers the potential income of the local population using available natural goods (reindeer pastures, water, rivers for fishing, areas for hunting, sites of vegetation for gathering berries, mushrooms, medicinal plants, etc.). In essence, it is proposed to determine *potential* incomes and losses, if this territory would be involved by the indigenous population in economic circulation (although this territory may not be used at the time of the study). The proposed method could be used in the case of temporary withdrawal of these territories for the project.

The authors used the indicators of resource productivity of the territory to determine losses to indigenous peoples in Yakutia. The analysis of ethnological expertise showed that many natural goods have no market value. The local population uses them for their own consumption. There are no established market prices for traditional products (meat, fish, wild plants, berries, and mushrooms), and standards have not been developed to characterize the biological productivity of land in different Arctic areas (the density of hunting resources per 1 ha, the productivity of fish per 1 ha, etc.). These and other methodological questions determined the need for the development of methodological approaches to calculate the losses. Other issues of methodology development include tools for geobotanical zoning of the territory, taking into account the allocation of exclusion zones and stress zones (indirectly, passively affected areas) [18].

Among basic sectors of traditional economy, domesticated reindeer herding is considered as the main type of economic activity preserving traditional culture. A special feature of this industry is the year-round grazing of animals by shepherds, which forces them to use their language, customs and traditions every day. Reindeers in the literal sense feed the indigenous peoples: they give the main food (meat), are used as a vehicle, provide traditional winter clothing or shoes and are an indispensable tool for the construction of yurts. Only in reindeer herding do the indigenous peoples not encounter competition from the dominant society. Hunting and fishing also play important roles for indigenous peoples and their communities. For example, "The Yukagir" tribal community—a hunter–fishermen aboriginal family from Ust-Yansky region in Yakutia—owns an area of 1914 thousand hectares. The are 17 members of the community, 15 of them men. They hunt, fish and mine mammoth tusks. The main sources of income for the community is commercial fish (*coregonus nasus, coregonus, and coregonus albula*) with a total volume of 52 tons, which is more than 5 million rubles [19].

Consider the case of losses' calculation of indigenous communities due to investment project in Anabar region. This project aims at the exploration and extraction of placer diamonds on the Polovinnaya River. The project is being developed by Arctic Capital LLC. The area is related to the territories of traditional nature use [20]. The indigenous communities perform reindeer herding, hunting and fishing there. The license area is located in the valley of the Polovinnaya riverbed and its tributaries (Figure 5). The total length of the study area is 75.4 km.

The project means partly and temporary traditional lands withdrawal. Therefore, the losses of reindeer herding, hunting, fishing, and gathering wild plants (berries, herbs, mushrooms, etc.) are the main indicators and defined based on the decreasing of traditional lands' productivity. The mine site has been divided into the exclusion zone and the stress zones. For the license period, it is assumed that biological resources are inaccessible for traditional nature use only in the exclusion zone, so their economic reserves are subject to compensation.

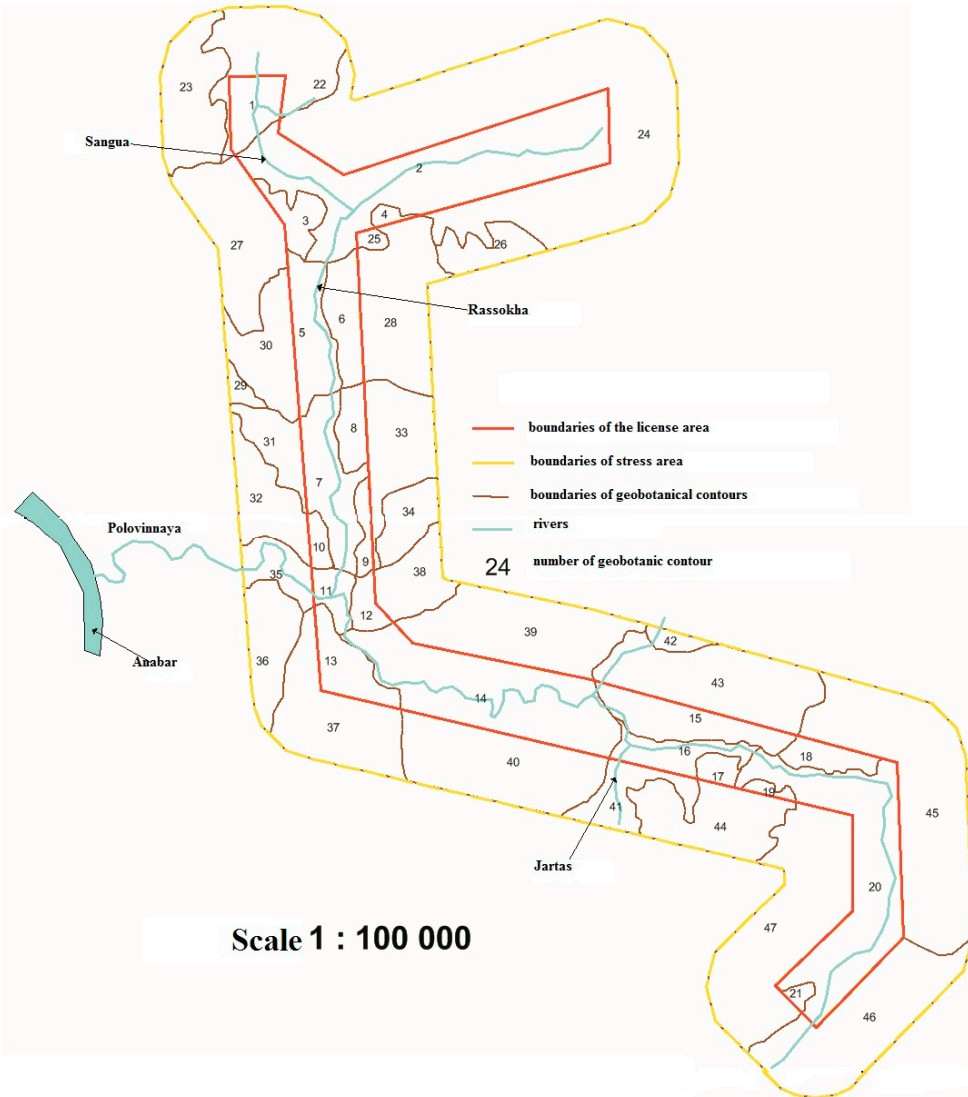

**Figure 5.** The license area of the project on Polovinnaya River, Anabar region, Yakutia.

## 4. Results

Losses of indigenous peoples are the lost profit. It is the gross annual income received from 1 ha. Annual gross income is defined as the difference between gross output of all traditional economic activity and their material and technical costs.

### 4.1. Reindeer Herding

Calculation of the losses of reindeer husbandry was carried out based on geobotanical mapping (Figure 5) [20]. All calculations were made by authors. On the map, for each geobotanical contour, the values of reindeer carrying capacity were determined. Based on the economic-geobotanical map of the mine area, its electronic version was created. The cost of reindeer herding production reduces with the reducing the reindeer carrying capacity of pastures due to industrial development [21]. It was assumed that traditional lands in exclusion zone completely lose their reindeer carrying capacity. In the stress zone, there is a partial decrease of this indicator. As a rule, the losses were calculated for one year [22]. On the economic-geobotanical map, the reindeer carrying capacity shows how many deer can graze on one hectare per day. Table 2 shows a fragment of the contour of the economic-geobotanical map. Table 3 shows a fragment of the calculation of losses in the research area (reindeer herding).

**Table 2.** Contour of the economic-geobotanical map of the license area at Polovinnaya River.

| Number of Geobotanical Contour | Zone | Main Geobotanical Species | Concomitant Geobotanical Species | Area, ha | Reindeer Carrying Capacity, Deer Per Day for 1 ha/winter | Reindeer Carrying Capacity, Deer Per Day for 1 ha/Summer |
|---|---|---|---|---|---|---|
| 19 | Exclusion | Lichen tundra | Hummock tundra | 276.7 | 4.9 | 0.0 |
| 20 | Exclusion | Tundra of cotton grass-lichen | Bushy-Lichen tundra | 2049.3 | 7.0 | 5.0 |
| 14 | Exclusion | Hummock tundra | Bushy tundra | 466.8 | 1.5 | 6.6 |
| 11 | Exclusion | Moss tundra | Tubercle tundra | 578.7 | 0.0 | 6.8 |
| 6 | Exclusion | Bushy tundra | Swamps polygonal-roller and fissured hilly lichen | 340.2 | 10.8 | 6.6 |
| 10 | Exclusion | Hummock tundra | Bushy-Lichen tundra | 110.4 | 5.3 | 5.8 |
| 27 | Exclusion | Lichen tundra | Spotty lichen tundra | 45.6 | 14.0 | 12.0 |
| 18 | Stress | Hummock tundra | Swamps polygonal-roller and fissured hilly lichen | 698.4 | 3.3 | 1.4 |
| 16 | Stress | Hummock-lichen tundra | Moss tundra | 902.1 | 1.8 | 1.5 |
| 21 | Stress | Bushy and lichen tundra | Tundra willow-lichen | 961.2 | 3.6 | 0.8 |
| 14 | Stress | Hummock tundra | Shrub tundra | 538.9 | 0.4 | 1.7 |
| 7 | Stress | Swamps polygonal roller and fissure hilly | Herbs | 270.9 | 0.0 | 1.6 |

**Table 3.** The calculation of losses in the license area (reindeer herding).

| Number of Geobotanical Contour | Zone | Area, ha | The Cost of the Gross Stock of Bioresources, Ruble/ha | Cost of Bioresource Production (Reindeer Herding), Ruble/ha | Financial and Technical Costs for the Year of Reindeer Herding, Ruble/ha | Gross Income of Reindeer Husbandry from 1 ha of Pasture (Contour), Ruble/Year | The Amount of Current Losses on the Contour, Ruble/Year |
|---|---|---|---|---|---|---|---|
| 19 | Exclusion | 276.7 | 328.68 | 75.60 | 3.56 | 72.03 | 19,931.48 |
| 20 | Exclusion | 2049.3 | 469.54 | 107.99 | 5.09 | 102.90 | 210,881.17 |
| 14 | Exclusion | 466.8 | 442.71 | 101.82 | 4.80 | 97.02 | 45,290.70 |
| 11 | Exclusion | 578.7 | 456.13 | 104.91 | 4.95 | 99.96 | 57,849.10 |
| 6 | Exclusion | 340.2 | 724.44 | 166.62 | 7.85 | 158.77 | 54,012.25 |
| 10 | Exclusion | 110.4 | 389.05 | 89.48 | 4.22 | 85.26 | 9413.07 |
| 9 | Exclusion | 90.1 | 368.93 | 84.85 | 4.00 | 80.85 | 7284.87 |
| 27 | Exclusion | 45.6 | 939.08 | 215.99 | 10.18 | 205.81 | 9384.85 |
| 18 | Stress | 698.4 | 223.03 | 51.30 | 2.42 | 48.88 | 34,137.37 |
| 16 | Stress | 902.1 | 120.74 | 27.77 | 1.31 | 26.46 | 23,870.49 |
| 21 | Stress | 961.2 | 239.80 | 55.15 | 2.60 | 52.55 | 50,515.43 |
| 14 | Stress | 538.9 | 110.68 | 25.46 | 1.20 | 24.26 | 13,071.53 |
| 7 | Stress | 270.9 | 105.65 | 24.30 | 1.15 | 23.15 | 6272.26 |

### 4.2. Hunting

Hunting is the second most important traditional activity of the indigenous peoples [23]. Local population hunts for their own consumption. This indicator is almost absent in official statistical data, as is fishing. The one method for real price determination is polling the local population. It was assumed that, during study period, the lands in the exclusion zone lose their value as hunting grounds. In the stress zone, due to the disturbing effects, the productivity of hunting grounds is reduced by 50%.

In Anabar region, the main object of hunting is wild reindeer [24]. The losses of the hunting industry due to land withdrawal for industrial purposes were determined based on the biological and economic reserves per 1000 ha of hunting lands. This approach was used for determination of hunting quotas for various animals (wild reindeer, white fox, elk, etc.). Table 4 presents information about the economic reserves of animals in the form of density indicators.

**Table 4.** Annual productivity average of hunting in Anabar region, Yakutia.

| Type of Animals (Birds) | The Population Density of This Species, Individuals Per 1000 Hectares |
| --- | --- |
| Wild reindeer | 0.214–1.680 |
| Elk | 0.353 |
| White fox | 0.370–0.540 |
| Ermine | 0.360–0.650 |
| Wolverine | 0.039 |
| Squirrel | 0.360 |
| Sable | 0.902 |
| Fox | 0.142 |
| Hare | 0.700–1.427 |
| Goose, individuals/10 km of the riverbed, lakeshore | 0.300–0.700 |
| Partridge | 0.836 |
| Wood grouse | 2.910 |
| Cock of the wood | 0.620 |
| Duck, individuals/10 km of the riverbed, lakeshore | 0.525 |

### 4.3. Fishing

The Polovinnaya River and its tributaries belong to the river basin of Anabar. Regular fishing is not peculiar here. Fishing quotas are not defined. According to ichthyologists, as a rule, the fish productivity of the Arctic rivers is 3 kg of fish per 1 ha of water surface. Since the catch is mainly dominated by relatively low-value (quota-free) fish species, the average price in the calculations was 300 rubles/kg. The amount of budget subsidies provided by local government is 30 rubles on 1 kg. The price for fish with subsidies is 330 rubles for 1 kg of fish. The overall reduction in catch volumes would be equal to the product of the fish productivity index per 1 ha of water surface over the entire water surface area within license area.

### 4.4. Wild Plants

Wild plants are harvested by the indigenous peoples for their own consumption. In conditions of limited sales, their prices do not exist. According to a survey of the population, average prices were determined for the main species of wild plants (lingonberries, blueberries, cloudberries, and mushrooms). The harvest reduction of wild plants occurs only in the exclusion zone and does not apply to the stress zone areas. The cost of economic reserves of wild plants is equal to the product of their economic reserve (kg) in the contour on the price of the wild plants (ruble/kg), Table 5.

**Table 5.** A fragment of the calculation of the value of economic reserves of wild plants in the exclusion zone, ruble.

| Number | Main Geobotanical Species | Concomitant Geobotanical Species | Area, ha | Cost of Potential Gross Output from the Whole Area of Contour, Rubles | | | | |
|---|---|---|---|---|---|---|---|---|
| | | | | Lingonberries | Blueberry | Cloudberry | Mushrooms | Total |
| 19 | Lichen tundra | Hummock tundra | 276.7 | 13,282 | 10,791 | 69,175 | 55,340 | 148,588 |
| 16 | Hummock tundra | Moss tundra | 208.2 | 4997 | 8120 | 31,230 | 31,230 | 75,577 |
| 21 | Hummock-lichen tundra | Bushy-Lichen tundra | 47.2 | 1699 | 1841 | 11,800 | 9440 | 24,780 |
| 11 | Moss tundra | Tubercle tundra | 578.7 | 20,833 | 22,569 | 86,805 | 86,805 | 217,013 |

The authors by experimental methods established that, during exploration and extraction, processes can affect up to 10% of the licensed area, i.e., the amount of possible losses of the traditional land could be 10% of the total losses all license area.

A special methodology for calculating of losses was developed by the Giprozem Research Institute. It was approved by the Ministry of Regional Development of the Russian Federation in 2009. Unfortunately, mistakes made in the development of the methodology led to negative results in its further application. It was based on the income method—the calculation of lost annual gross income of stakeholders as a result of traditional lands withdrawal, Table 6. However, it does not allow calculating the damage to intangible heritage—the language and the original culture of Aboriginal peoples. These issues of calculating of losses were left outside the legal regulation, the provision of new land plots was withdrawn, and status issues that were indirectly influenced by the industrial development of the territory were ignored. Therefore, a federal law on ethnological expertise, which is currently actively discussed in the State Duma, is needed to extend the experience of Yakutia to other regions. It would help to defend indigenous peoples' rights in Russia [25].

**Table 6.** Fragment of the calculation of losses (wild plants) [20].

| Number of Geobotanical Contour | Area, ha | Gross Product Losses from the Whole Contour Area, Ruble/Year | Material Costs for the Collection of Wild Plants, Ruble/Year | Current Losses, Ruble/Year |
|---|---|---|---|---|
| 19 | 276.7 | 14,859 | 700 | 14,158 |
| 20 | 2049.3 | 143,451 | 6762 | 136,689 |
| 14 | 466.8 | 40,285 | 1899 | 38,386 |
| 11 | 578.7 | 21,701 | 1023 | 20,678 |
| 6 | 340.2 | 17,861 | 842 | 17,019 |

Existing approaches to calculate of losses do not consider the social aspects of the project impact. In our opinion, industrial companies acting in the territories of traditional nature use should also compensate for social damage to improve the quality of life of the indigenous communities (construction of social facilities, financing of ethno-cultural events, employment) [26]. As practice shows, calculating the losses to indigenous peoples may differ significantly due to methodology imperfections. According to currently using methodology, possible losses of the indigenous peoples should be calculated using the coefficient of conversion of the lost annual gross income into loss of profit, considering the time period of recovery of the disturbed area. However, traditional production cannot be restored unless disturbed natural resources are restored [27]. The existing methodology does not reflect questions about the recipient of compensation (communities, local governments, and public organizations). In addition, the questions need to be adjusted for determination the areas of stressful impact of the project as they move away from the industrial facility. These important aspects of the imperfection of the methodology should be resolved, since legal gaps do not make it possible to determine a fair

amount of compensation to indigenous peoples [28]. The authors' approach, based on the resource productivity of traditional lands, allows calculating losses to indigenous peoples. The authors suggest adding the results with the recommendations on the use of a compensation fund to socio-economic and environmental problem-solving.

## 5. Discussion

To calculate losses, it is advisable to apply an income approach in terms of lost profits. The lost profit of land users depends primarily on the area (radius) of technogenic impact. Technogenic impact is the impact of industrial technology, transport and communications that can cause ecosystem disruption. It varies in duration (short-term, long-term, and cyclical), degree (super-weak, weak, strong, and super-strong), admissibility (permissible and unacceptable), and controllability (controlled and uncontrolled). To assess the intensity of the industrial impact on natural complexes, it is proposed to take into account such factors as the hazard class of the project, the nature of violations (areal—quarries and other industrial facilities; linear—pipelines, highways, etc.). The hazard class is a conventional value intended for the simplified classification of potentially hazardous substances on the basis of data on the toxicological properties. Industrial impact assessment involves the detailed definition of technologies and chemicals used in mining and processing. In our opinion, methods for calculating of the losses of traditional nature use are based on indicators of income decline from traditional activities. They should be complemented by the environmental and socio-economic costs that are additionally borne by subsoil users to improve in livelihoods of indigenous peoples (job creation, spending on education, lifestyle changes, etc.). One of the main features of economic development projects of traditional lands is their multi-criteria character. Many different socio-economic groups, local governments, business, NGO and indigenous peoples are engaged in exploration and extraction processes. At the same time, commercial efficiency, which usually prevails the social effect of the project, cannot always be used to evaluate the benefits of alternative projects. In projects when implementation results may have implications affecting social, environmental, ethnological, cultural and other aspects of indigenous livelihood, such criteria become important. In the context of multi-criteria strategies for the economic development of traditional lands, differences in the importance of individual criteria for each of stakeholders whose interests are affected by the project, the substantiation of a rational option for the development of such areas is significantly complicated. An obvious principle when evaluating strategies for the economic development of traditional lands is their profitability and consistency of interests of the main parties. In this case, the profitability cannot be unambiguously determined only by the ratio of economic results and costs associated with their implementation. This concept should include the degree of satisfaction of the interests of various parties, which depend on the implementation of such strategies. In turn, the consistency of interests can be achieved only if the results of the implementation of strategies are "beneficial" for all parties whose goals and interests may not coincide. To mitigate the negative social consequences of industrial development in the traditional lands and economic activities of indigenous communities, it is necessary to conclude a tripartite agreement on cooperation and financing of specific programs for promoting sustainable development. Such agreement could be concluded between the investors, local authorities and indigenous peoples. When drafting laws at the federal level on benefit sharing and procedure for compensating damage to the indigenous peoples, it is necessary to consider the experience of Yakutia where there are already successful examples of the agreements on cooperation and partnership realized by the largest mining companies in Russia. The fair compensation provides the sustainable development and helps all stakeholders to avoid conflicts and preserve unique ethnological heritage of the indigenous peoples of the North.

**Author Contributions:** I.P. conceptualized the presented idea, prepared original draft and developed a formal analysis. V.G. investigated and managed the resources of the research; review & editing are also her responsibility. All authors discussed the results and contributed to the final manuscript.

**Funding:** This research was funded by Russian Foundation for Basic Research, grant numbers 19-010-00023a.

**Conflicts of Interest:** The authors declare no conflict of interest.

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
