# Peer review of "The Compensation for Losses to Indigenous Peoples Due to the Arctic Industrial Development in Benefit Sharing Paradigm"

_resources, doi:10.3390/resources8020071_

Reviewer 1 Report

I'm still not quite pleased with the language, even if it has improved a lot! That's great! I will not have time to point out problematic parts/sentences this time. But I think that it still needs to be worked at. I feel sorry to say that.

There's still one 'habitat' left there. 

1. Introduction: a very long chapter. Subtitles? 

The list of the "Key points of the article" (line 105 =>) is still a bit strange. Why there must be this list with numbers? And I don't get the point of it: there are aims, methods, results... Like an extended abstract?

2. The study area: is all that info really necessary or relevant, e.g. temperature and mortality in the region??

3. Materials and Methods: you talk e.g. about concept of benefit sharing, and not about materials? Check the content vs. title.

Section 3.1 Research concept: too long.

5. Disucssion: is technogenic impact (why a new concept in conclusions?? perhaps it's a common word, but it's unfamiliar to me.) same as industrial impact? Also 'hazard class' is a new concept here, and not quite understanble.

Author Response

We sincerely thank the Reviewer for the deep and attentive altitude to our manuscript.   

1. We include Convention to the text and made reference on it (5). The Convention is mentioned in line 71.

2. The compensation term is briefly explained in line 76.

3.  “Habitat” is changed to traditional lands.

4. We decided to delete key points of the paper. We agree that they are superfluous in fact.

5. We delete general climate and geographic information in 2. The study area. However, we believe that the socio-demographic indicators of communities are relevant, as they give the reader an idea of the state and characteristics of the object of study, its main development trends. We would like this information to be kept.

6.  Technogenic impact and hazard class are terms using in the industrial impact assessment. We added their brief explanations.

7. We changed the title of paragraph 3 to 3. Concepts and Methods, and 3.1 to 3.1 Benefit sharing concept in Russia

8. We request the English editing service.

Reviewer 2 Report

As a general remark, it may be interesting to explain the concept of compensation adopted here: this is compensation in the sense of damages for wrongful behaviour or just a fairer distribution of social costs by way of public regulation forcing the companies to share their benefits with the indigenous peoples? Why is tort law not used instead? 

Apart from that, it may strike the reader that no attention is paid to ILO Convention no.169. Is it not relevant to the subject matter of this paper? (please see https://www.ilo.org/dyn/normlex/en/f?p=NORMLEXPUB:12100:0::NO::P12100_ILO_CODE:C169).

The abstract point out that the foreign experience is also studied, but the paper mainly revolves on the Russian situation. 

As to some minor, formal issues: 

There is a comma too much in line 106 ("article,:") and a closing parentheses too few in line 138. The map after line 162 has been prepared for the occasion (or are there copyright issues here?). No reference is provided about the source or permission to use it. When it is said that most deaths are due to alcohol use (line 184), some further explanation would be welcome (is this because of the bad economic situation? Is there any study which can be cited here?). 

As regards the section on Research concept, it may seem a bit strange that the purpose of the article is presented here (lines 358, 370, ff.) instead of at the very beginning of the paper. It is suggested that the section starts with the explanation provided in line 373 instead, properly readjusted. Apart from that, please elaborate a little further on the imperfection of the ownership system among indigenous peoples: it is due to practical issues, lack of land registry, insufficient protection, ...? (line 408). Please reconsider this sentence: "Indigenous peoples daily use native language, traditions and customs" (lines 440-441). Please explain the reference to the "experts" in line 520 (a source or note may be needed). The methodology referred to in line 531, has it been published? Why is Agreement written with initial capital letter in line 588? (compare to the following line). 

Author Response

We sincerely thank the Reviewer for the deep and attentive altitude to our manuscript.   

1. The paper is devoted mainly for the Russian cases in benefit sharing and compensation tools as a part of this concept. We study foreign theory and practice but discuss mostly the opportunity for improvement of the economic rights of indigenous peoples in Russia.

2. We replaced the purpose of the article to the beginning (line 44).

3. The web-reference to the map was added.

4. Cause of alcohol addiction of the population is explained in the line 151.

5. In line 329 the problem of land property is described.

6. In line 422 the phrase about indigenous peoples and their language was changed.

7. Line 503: “experts” were changed on “the authors by the experimental methods”.

8. Line 514-517 sentences on methodology were rephrased.

9. We still request the English editing service